# The Use of the FEM to Identify the Optimal Groove Dimensions Ensuring the Least Stressed Connection between a Zirconia Coping and Veneering Ceramic

**DOI:** 10.3390/ma11122360

**Published:** 2018-11-23

**Authors:** Beata Śmielak, Leszek Klimek, Jacek Świniarski

**Affiliations:** 1Department of Dental Prosthetics, Medical University of Lodz, ul. Pomorska 251, 92-213 Lodz, Poland; 2Department of Materials Research, Institute of Materials Science and Engineering, University of Technology, ul. Stefanowskiego 1/15, 90-924 Lodz, Poland; leszek.klimek@p.lodz.pl; 3Department of Strength of Materials, University of Technology, ul. Stefanowskiego 1/15, 90-924 Lodz, Poland; jacek.swiniarski@p.lodz.pl

**Keywords:** FEM, zirconia, veneering ceramic

## Abstract

**Background:** To examine the influence of coping notches with varying groove widths and depths on the quality of the connection with ceramic. **Methods:** Ten rectangular sintered zirconia (3Y-TZP) samples were etched with a neodymium-yag laser Nd:YAG. Then, a profilometer was used to test the depths and spacing of the grooves. A notch profile was used to design the shapes and spacing of the grooves based on a finite element method (FEM) simulating zirconia. The following situations were simulated: an increase in groove width from 100% to 180% and depth from 40% and 80%; and a 40% depth and width. **Results:** An increase of 10% in the baseline width caused an insignificant reduction of the strain in the connection. A further increase in this dimension led to a 50% increase in strain with a 40% increase in width. An increase in the groove depth by 40% reduced the strain level by 13%, while an increase in the groove depth by 80% reduced the strain level by 22%. Simultaneous deepening and widening of the groove by 40% had no significant impact on the strain level. **Conclusion:** Maintaining the width of the groove bottom while increasing the depth offers fewer advantages than deepening and narrowing the groove bottom.

## 1. Introduction

The basic advantages of restorations on a zirconia base are biocompatibility, perfect aesthetics, the marginal integrity of the structure, color stability, and low bacterial adhesion [1,2,3,4,5,6,7]. The main complication of such restorations is damage to the veneering porcelain [8,9,10,11,12,13,14,15]. Damage tests show, in a large number of cases, chipping of the veneering porcelain (15–62%), cracks (25–50%), delamination (>10.7%), and large fractures (3–33%). Failures, mainly of an adhesive nature, occur more often than in metal – ceramic restorations [16,17]. An insufficient connection is the main cause of the failures.

The quality of the connection between zirconia and the veneering ceramic depends on three factors: chemical-diffusion bonding, the connection resulting from the difference in shrinkage of both materials, and the mechanical connection of the microattachments formed as a result of the penetration of the liquid ceramic into the uneven surface of the substructure. The chemical-diffusion bonding mechanism has not been clearly elucidated [18,19]. It probably involves the mutual dissolution of the ceramic and zirconia. Selecting proper thermal expansion coefficients (α) is absolutely vital for achieving a good connection between the veneering ceramic and zirconia. To minimize the tensile stresses in the ceramic, the veneering material should have a value of α equal to or slightly less than that of the coping material [12]. Achieving the right thermal expansion coefficient is possible, but other factors connected with ensuring even adherence and adhesion between the zirconia and the ceramic are vital for a successful outcome. 

The majority of authors are of the opinion that the quality of the connection between the ceramic and zirconia is determined by a suitably developed coping surface [20,21]. The treatment of the zirconium core can be divided into mechanical and chemical methods. Chemical methods usually involve the application of hydrofluoric acid (HF). However, the considerable hardness of the zirconia surface ensures only minimal roughness on the nanoscale, even after using an experimental 40% acid solution [22].

The most widely used mechanical method is abrasive blasting with aluminum oxide. Although many papers have been published describing its effect on surface development, depending on the grain size (25 to 250 μm), the amount of pressure (1.5–3 bars), the treatment time (10–20 s), and the distance of the nozzle from the sandblasted samples, there are no clear recommendations in this area [23,24]. Despite many studies in this area, no effective models have been devised for properly preparing the substructure of the crowns or bridges to ensure a good connection with the veneering ceramic, which is essential for ensuring long-term clinical success [13,20,25,26,27,28,29,30,31,32,33].

According to a study conducted by Fischer et al., the quality of the connection between ceramic and zirconia depends on the mechanical anchoring of the ceramic and the zirconia, but also on the types and concentrations of defects on the surface, as well as the stress level in the ceramic layer [13]. Hence, it is important to determine how effective different treatment parameters are in achieving optimal conditions of the connection. What is known is that zirconia surfaces should be shaped in such a way that any resulting irregularities possess appropriate dimensions and shapes and that the veneering material connects homogenously with the coping. 

Designing the ideal surface development conditions to select the right laser beam parameters and checking the quality of the connection after applying the appropriate forces is possible thanks to the finite element method (FEM). FEM is a modern technique that is currently used on a very wide scale in scientific research on biomechanics. 

This method is also employed in dentistry. We utilized it to perform analyses of stress and deformation in tooth structures, in teeth restored with simple fillings, post-and-cores, in prosthetic crowns, prosthetic substrates under uni- and bilateral bridges, in teeth missing periodontal support, in peri-implant bone, and in new ceramic materials [34,35,36,37,38,39].

The idea of the finite element method (FEM) is to replace the continuous medium of the tested object with a system of smaller parts called finite elements. These elements are only connected to each other at points called nodes. In the mathematical theory of FEM, ‘shape functions’ are introduced, which ensure the continuity of the construction. This means that following deformation, the edges of adjacent elements are in close proximity to each other. Each node has up to six degrees of freedom. Boundary conditions in the form of forces, displacements, and temperature fields are only applied in the nodes. The condition of equilibrium must be fulfilled for each node. Node displacements form a system of unknowns which can be calculated when the loads are given. Node displacements are basic variables that make it possible to calculate the physical values required for any presentation of results. 

In the absence of studies on the influence of a roughness profile shape on the quality of the connection between a coping and a fired ceramic, an attempt has been made to address the problem and identify the phenomena-taking place at the interface of both materials. In addition, the impact of different coping notch depths and heights on the connection strain has been studied.

The aim of these calculations was to determine the pressures between the zirconia coping layer and the veneering ceramic, as well as the shear stresses of the connection, in the selected variants simulating groove size.

## 2. Material and Methods

Ten rectangular 3Y TZP Ceramill Zi (Amann Girrbach AG, Koblach, Austria) plates were sintered in a furnace (Ceramill Therm; Amann Girrbach AG, Koblach, Austria) using a universal program (8°/min from 200° to 1450° for 2 h at a constant temperature of 1450° and with a suitable cooling time). The sintering process lasted approximately 10 h. Material shrinkage amounted to approximately 21%. Following sinterization, the plates had the following dimensions: 10 × 10 × 5 mm^3^. 

To determine the initial treatment parameters, the surfaces of the samples were etched unidirectionally with a YAG Nd laser (Fidelis, Fotona, Ljubljana, Slovenia). The use of a beam with a wavelength of 1070 nm produced an average power of 6–20 W. The duration of the impulse was 30–200 ns, the scanning speed was 100–600 mm/s, and the impulse frequency was 25–125 kHz. One randomly selected plate was chosen for further tests. Figure 1 shows the shape and distribution of the notches. 

Then, the depth and spacing of the grooves were examined with a profilometer. 

The dimension characteristics for A, B, C, D, E, F, and G were obtained after measuring the profile of the notches made in Figure 1 and Figure 2, and are summarized in Table 1. 

The notch profile shown in Figure 2 was used to design the shape and distribution of the grooves on an FEM model simulating a zirconia coping, as shown in Figure 3. The grooves were made in one direction, horizontal to the load. The blue color denotes the zirconia coping and the green color represents the ceramic.

The mechanical properties of the materials adopted for calculation purposes are presented in Table 2.

The task was solved using the FEM Ansys v16.2 software system (ANSYS, Inc., Washington, PA, USA). Discretization was performed with the solid element SOLID186. The SOLID186 element is a twenty-node cuboid element with three degrees of freedom at each node. The model comprised 38,000 elements and 171,500 nodes. Placed between the base and the ceramic was a GLU-type contact based on TARGE170 and CONTA174 elements. Using an established contact made it possible to determine the contact forces and the shear forces between the elements in contact. In addition, there was no displacement between the contact pairs in the numerical task. The task was solved using large nonlinearities. 

The following situations were simulated:

An increase in the groove width in relation to the width measurement D (0.03 mm) from 100% to 180% at a frequency of 10%, with the other geometrical dimensions remaining constant.

An increase in the groove depth by 40% and 80% in relation to depth A (0.017) with width D (0.03 mm) and other geometrical measurements remaining constant.

An increase of 40% in the groove depth in relation to depth A (0.017 mm) and a 40% increase in the groove width in relation to width D (0.03 mm), with the other geometrical dimensions remaining constant.

## 3. Results

The results were presented in the form of color maps showing the normal, shear, and reduced stress σred according to the von Mises hypothesis. The results indicate that the strain level in the zirconia and ceramic coping should be treated in a qualitative rather than a quantitative sense. The color code, ranging from navy blue to red in the computer print-out legend, reflects the increase in the stress values. Identical coloring in a given area of the mathematical model indicates approximately the same physical force value at a given moment.

Figure 4 shows the pressure distribution on the coping surface at groove width D (0.03), where the other geometrical dimensions shown in Table 1 remain constant. When the edge of the first group is enlarged, the largest equivalent stresses are 45 MPa and are located at the edge of the groove.

Figure 5 shows the calculation results for a groove width on a zirconia coping that is increased at a frequency of 10%, from 100% to 180% of width D (0.03), with the other geometric dimensions remaining constant. 

An increase of 10% in the baseline width (0.03 mm) results in an insignificant reduction in the connection strain down to 43 MPa. A further increase in the width by 40% increased the strain by 50%. The connection strain declines once more to 45 MPa when the groove width is increased by 50% before slowly rising to 56 MPa when the width is increased by 70%. The stress then falls to 48 MPa, when the width is 80% greater, which is comparable in value to the stresses observed prior to the change in the geometrical dimension. 

Figure 6 shows the pressure distribution on the coping surface after a 40% increase in the groove depth in relation to dimension A (0.017) and with the initial groove width D (0.03). The angle of inclination of the groove walls remained constant. 

An increase in the groove depth results in a decline in the strain from 45 MPa to 39 MPa, which represents a reduction of 13%. Further increasing the groove depth is thus justified. 

Figure 7 shows the pressure distribution on the coping surface with a further 80% increase in groove depth in relation to dimension A (0.017), with the initial groove width D remaining constant at 0.03. The angle of inclination of the groove walls was also constant in this case. 

As can be observed here, a further increase in the groove depth, even a twofold increase, does not lower the strain of the material, and the values remain comparable to those noted when the depth is increased by 40%. 

In a further analysis, a model was simulated based on the basis of depth A (0.017 mm) and a 40% increase in the groove width in relation to width D (0.03 mm), with other geometrical dimensions remaining constant (Figure 8).

A simultaneous 40% increase in both the width and depth did not produce many benefits. Indeed, there is some at the connection, but was significantly less than when the groove depth alone was increased, which amounts to 41 MPa.

Based on the above analyses, it can be concluded that an increase in the groove depth, with other geometrical dimensions remaining constant, allows for a reduction in the connection strain. A 40% increase in the groove depth results in a 13% reduction in the strain level. A further 40% increase in depth, bringing the total increase in depth up to 80%, reduces the strain by 22%. Maintaining the width of the groove bottom while increasing the depth offers fewer advantages than deepening and narrowing the groove bottom. The gain in strain declines from 13% to just under 9%.

## 4. Discussion

Strength testing of teeth cannot be performed in the oral cavities of patients, since it could damage tissue. FEM enables the testing of structures with more complex shapes, composed of different materials and subjected to any load. The tests were carried out on computer models of the studied samples. The accuracy of the calculations is conditioned by the number of elements. To achieve credible results, the model should be divided into a large number of elements, which simultaneously increases the numbers of degrees of freedom, and in this way, significantly extends the computation time [40]. 

In the FEM tests, it was assumed that the connection between the individual materials was perfect and remained undamaged despite the increase in load. In reality, there is no such perfect connection. This is due to inaccuracies in the connections made in laboratory conditions [41].

The aim of the calculations was to determine the pressures between the layer of the zirconia coping and the veneering ceramic, as well as to estimate the shear stresses between the materials. The notches on the coping surface and the placement of the veneering ceramic on these notches were designed to relieve the strain on the adhesion connections of the ceramic and the coping. The aim was to support a mechanical shift in the load so that the connection between both materials was neither purely adhesive nor mechanical-adhesive in character. The grooves on the zirconium coping should have vertical walls, which would be the most advantageous for a connection. However, the laser technology used to make the grooves also at the same time makes it impossible to construct vertical walls. It likewise restricts their sizes and depths. 

Since it is impossible to make a clean mechanical connection, the inclined walls are loaded with surface pressures resulting from the shape of the connection and with the shear stress of the connection between the ceramic and the coping. A flat connection is known to have only the strength of an adhesive connection. Mixed adhesive-mechanical connections partly eliminate the risk of the adhesive connections breaking due to a shift in loads via the “latches” formed by the notches on the coping surface. 

However, in the corners of the grooves, there is a tendency for stresses to accumulate from the transferred mechanical loads. Areas of accumulation or concentration of the stresses or strains in the material are very small. Considering the actual surface rather than an ideal one, we can assume that these concentrations will be blurred by mechanical microlatches, which can be observed in Figure 2, in particular.

It is important to stress, however, that the results of the calculations are qualitative and not quantitative in character and require observation under experimental conditions. Not all aspects can be modeled within the framework of FEM calculations. In particular, there is no table available with critical stress values for the adhesion phenomenon.

## 5. Conclusions

The originally proposed nominal notch dimensions have been shown to be suitable due to the strain level in the connection.An increase in groove depth, with the other geometrical dimensions remaining constant, enables a reduction in the connection strain.A simultaneous 40% increase in the width and a 40% increase in depth does not significantly reduce the connection strain.This analysis may be useful when selecting the correct laser light beam parameters for zirconium surface conditioning.

## Figures and Tables

**Figure 1 materials-11-02360-f001:**
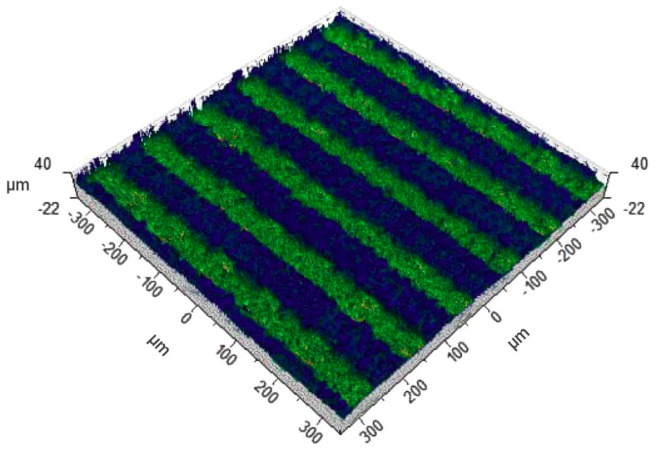
Shape and distribution of the notches on a sample zirconia surface.

**Figure 2 materials-11-02360-f002:**
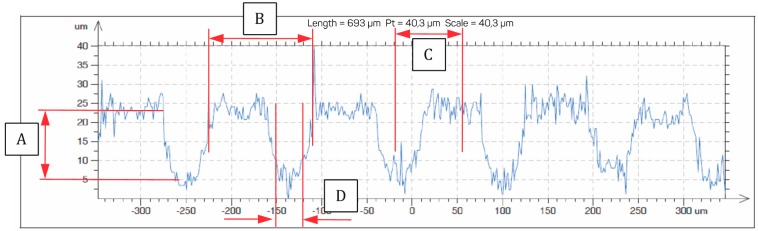
Notch profile on a sample zirconia surface.

**Figure 3 materials-11-02360-f003:**
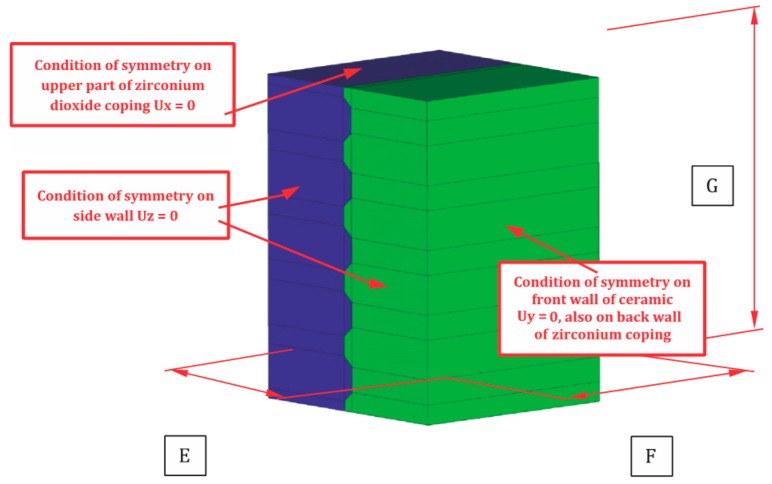
Shape and distribution of grooves, as well as the marginal conditions adopted for calculations based on the FEM model.

**Figure 4 materials-11-02360-f004:**
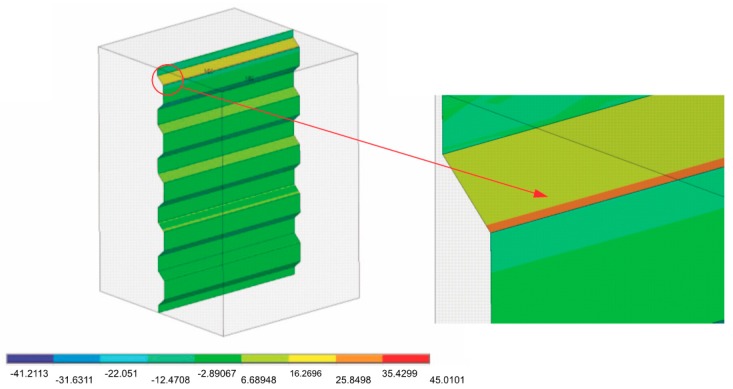
Pressure distribution on a coping surface and an enlarged image of the edge of the first groove. σcont max = 45 MPa.

**Figure 5 materials-11-02360-f005:**
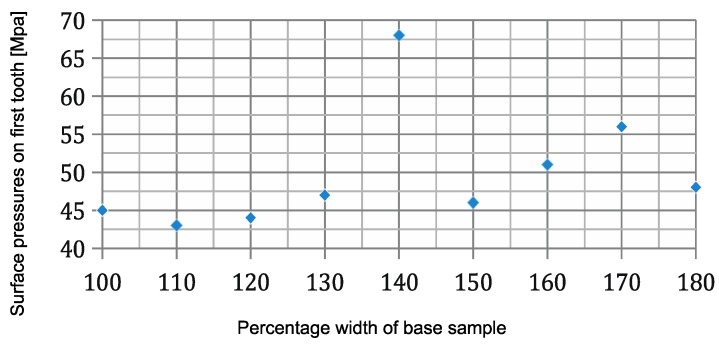
Surface pressure values on the first notch, depending on sample width.

**Figure 6 materials-11-02360-f006:**
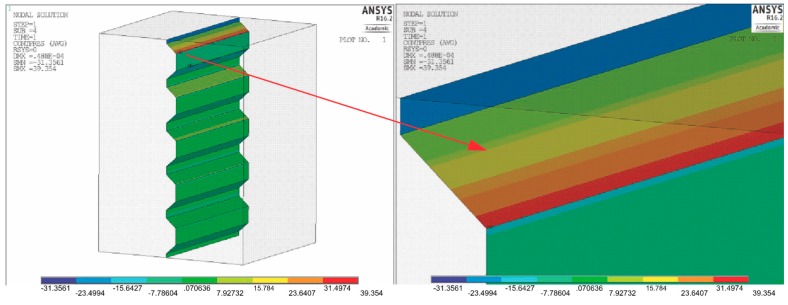
Pressure distribution on the coping surfaces and a magnification of the surface of the first groove. σcont max = 39 MPa.

**Figure 7 materials-11-02360-f007:**
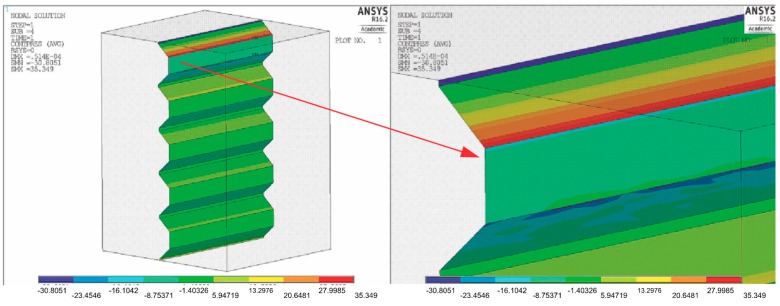
Pressure distribution of coping surfaces and magnification of the surface of the first groove. σcont max = 35 MPa.

**Figure 8 materials-11-02360-f008:**
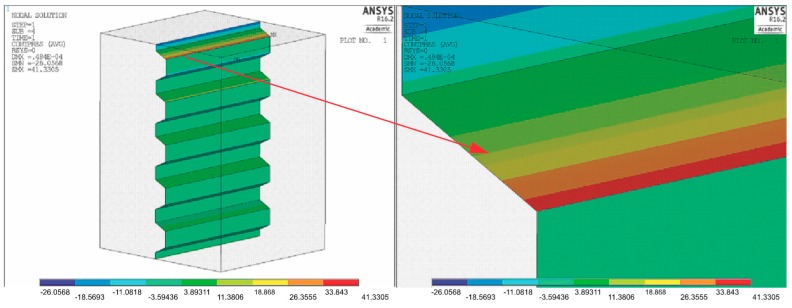
Pressure distribution of the coping surfaces and magnification of the surface of the first groove. σcont max = 41 MPa.

**Table 1 materials-11-02360-t001:** Dimension characteristics for A, B, C, D, E, F, and G adopted for the tests.

Measurement	A	B	C	D	E	F	G
Value (mm)	0.017	0.115	0.075	0.03	0.42	0.4	0.575

**Table 2 materials-11-02360-t002:** The material data for zirconia and ceramic adopted for calculation purposes.

Material	Young’s Modulus	Poisson Number
Zirconia	205 GPa	0.16
Ceramic	70 GPa	0.19

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
