# Peer review of "The Use of the FEM to Identify the Optimal Groove Dimensions Ensuring the Least Stressed Connection between a Zirconia Coping and Veneering Ceramic"

_materials, 2018, doi:10.3390/ma11122360_

Round 1
Reviewer 1 Report
This manuscript, entitled 'The use of the FEM to identify the optimal connection conditions for a zirconia coping and veneering ceramic,' is considered to be within the scope of this journal. The reviewer thinks that this manuscript is well organised. However, there are some issues to be addressed before it is acceptable for publication.
The title of this manuscript is about finding the optimal connection conditions. However, the reviewer could not find the optimal conditions for the connection between zirconia and veneering ceramic, based on the results of this study. The reviewer was just able to accept that the groove depth is more contributor to the reinforced binding than the groove width. The title is considered to be changed.
Although this manuscript mentions the limitations of this study, a real test is necessary to show the superiority of the authors' FEM model. A specimen with a real tooth shape, 0.5 mm thick zirconia core, and 0.5 mm thick veneering ceramic could show a very different load distribution pattern from that in this FEM study. At least, such a comparison between the FEM model and the real prosthodontic crown is needed.
Author Response
Dear Reviewer,
Thank you for your comments and suggestions concerning our manuscript.They are certainly appropriate.
1. Preparing a more comprehensive paper reporting the results from our study on zirconium dioxide, we did not realize that this part of the research was not appropriately correlated with the title.The title has been changed according to your suggestion.
2.It’s true that a specimen (as a whole) with a true tooth shape, 0.5 mm thick zirconium oxide core and 0.5 mm thick veneering ceramic, would exhibit different stress distribution compared to the specimens used in our FEM study. However, it was not our aim to analyse the stress distribution in the whole element, but only to check how the connection between the zirconium oxide substructure and veneer ceramics behaves, depending on the size of the grooves, and the FEM simulations were designed to fulfil this requirement. When loading a real prosthetic restoration, the shear component of the applied force would be responsible for the destruction of the connection, therefore the FEM was designed in such a way that the boundary conditions for pure shear destruction of the connection between veneering ceramic and zirconium oxide substructure were met. When we analyse the effect of pure shearing force on the connection between zirconium oxide and ceramic, the dimensions of the specimen do not matter, because the separation caused by shear forces occurs in a two-dimensional horizontal plane.

Reviewer 2 Report
The manuscript is well done and performed. The review falls into the MATERIALS aim and scope. However the topic is far and for this reason some more chapter or paragrpah should be added in the introduction section about Digital Dentistry or Cad Cam or Compouter assisted prosthodnotics and FEM analysis. At this stage the paper seems direct to the scientist leaving the clinical possibility or application of the study. For this reason it should be suggested to added some references in the introduction section in order to increase the value of the paper.
Overall a good paper please modify accordingly
FEM and Von Mises Analysis on Prosthetic Crowns Structural Elements: Evaluation of Different Applied Materials. Bramanti E, Cervino G, Lauritano F, Fiorillo L, D'Amico C, Sambataro S, Denaro D, Famà F, Ierardo G, Polimeni A, Cicciù M. ScientificWorldJournal. 2017;2017:1029574. doi: 10.1155/2017/1029574. Epub 2017 Apr 3.
Cervino G, Romeo U, Lauritano F, et al. Fem and Von Mises Analysis of OSSTEM ® Dental Implant Structural Components: Evaluation of Different Direction Dynamic Loads. The Open Dentistry Journal. 2018;12:219-229. doi:10.2174/1874210601812010219.
All those manuscripts should be added in the introduction section adding a portion about the clinical application of the FEM and Von mises study
Author Response
Dear Reviewer,
Thank you for your comments and suggestions concerning our manuscript.
They will certainly have a positive impact on the substantive value of the article.
As suggested, the introductory part has been extended to include general principles for FEM studies.
Literature based examples of FEM applications in dentistry have also been included.

Round 2
Reviewer 1 Report
The reviewer understands what the authors mean. The reviewer has nothing special to tell about this revised version of the manuscript.
Reviewer 2 Report
Authors made excellent job addressing all the reviewer comments and requests